# Fetal Surgery for Gastroschisis—A Review with Emphasis on Minimally Invasive Procedures

**DOI:** 10.3390/children9030416

**Published:** 2022-03-15

**Authors:** Lidya-Olgu Durmaz, Susanne Eva Brunner, Andreas Meinzer, Thomas Franz Krebs, Robert Bergholz

**Affiliations:** 1Department of General, Visceral, Thoracic, Transplant and Pediatric Surgery, University Medical Center Schleswig-Holstein (UKSH), Kiel Campus, Arnold-Heller-Strasse 3, 24105 Kiel, Germany; olgu.durmaz@outlook.de (L.-O.D.); thewalkingsmad@gmail.com (S.E.B.); andreas.meinzer@uksh.de (A.M.); thomas.krebs@kispisg.ch (T.F.K.); 2Department of Pediatric Surgery, Children’s Hospital of Eastern Switzerland, Claudiusstrasse 6, 9006 St. Gallen, Switzerland

**Keywords:** prenatal surgery, minimally invasive surgery, fetoscopy surgery, open fetal surgery, in utero intervention, EXIT, amnioinfusion, amnio exchange, complex gastroschisis

## Abstract

(1) Background: The morbidity of gastroschisis is defined by exposure of unprotected intestines to the amniotic fluid leading to inflammatory damage and consecutive intestinal dysmotility, the viscero-abdominal disproportion which results in an abdomen too small to incorporate the herniated and often swollen intestine, and by associated pathologies, such as in complex gastroschisis. To prevent intestinal damage and to provide for growth of the abdominal cavity, fetal interventions such as amnio exchange, gastroschisis repair or covering have been evaluated in several animal models and human trials. This review aims to evaluate the reported techniques for the fetal treatment of gastroschisis by focusing on minimally invasive procedures. (2) Methods: We conducted a systematic database search, quality assessment and analyzed relevant articles which evaluate or describe surgical techniques for the prenatal surgical management of gastroschisis in animal models or human application. (3) Results: Of 96 identified reports, 42 eligible studies were included. Fetal interventions for gastroschisis in humans are only reported for EXIT procedures and amnio exchange. In animal models, particularly in the fetal sheep model, several techniques of open or minimally invasive repair of gastroschisis or covering the intestine have been described, with fetoscopic covering being the most encouraging. (4) Discussion: Although some promising minimally invasive techniques have been demonstrated in human application and animal models, most of them are still associated with relevant fetal morbidity and mortality and barely appear to be currently applicable in humans. Further research on specific procedures, instruments and materials is needed before any human application.

## 1. Introduction

Gastroschisis is a congenital malformation consisting of a right-sided paraumbilical defect of the abdominal wall through which the intestines herniate from the fetal abdomen into the amniotic cavity. Its prevalence is increasing for reasons currently unknown, reaching nearly 4 cases per 10,000 live births [1]. An association with low maternal socioeconomic status, early maternal age, and low preconceptionally body mass index is suggested [2]. Environmental and toxic substances such as medications, smoking, drugs, and pollution also appear to have an impact on its pathogenesis [3,4,5].

### 1.1. Pathology

Gastroschisis results in distinct pathological changes of the herniated intestines and the fetus itself, which may lead, not only to their accumulation, but also to the characteristic complications and morbidity of this malformation.

First, in contrast to omphalocele, the herniated organs are not covered by the umbilical cord or peritoneal membranes [6]. Thus, exposure of the uncovered intestine to the amniotic fluid affects the intestinal wall itself which may lead to the development of an inflammatory peel around the intestinal loops. This induces their rigidification and thickening, making it impossible to identify the bowel loops separately [7]. The distinct pathomechanism appears to be the effect of the exposure of many molecular components of the amniotic fluid onto the exposed intestine, including interleukins, ferritin, bile components, and digestive enzymes [8,9,10,11].

Second, the extra abdominal growth of the intestines reduces the relative volume of the abdominal cavity. This viscero-abdominal disproportion often impairs postnatal reposition of the intestines into an abdominal cavity being too small, which causes an increased risk for an abdominal compartment syndrome after forced primary closure of the defect [12].

Third, even after successful postnatal surgery of gastroschisis, further disease progression is influenced by the frequently occurring intestinal transport disorder. The pathogenesis of this intestinal dysmotility in gastroschisis is not completely understood. It is hypothesized to be the result of a damaged enteric nervous system and the deficiency of interstitial cells of Cajal (ICC). These pacemaker cells of the gastrointestinal tract control the contraction of the smooth muscles and thus the rhythmic pattern of peristaltic waves [13,14,15,16]. Long exposure of the intestine to the amniotic fluid causes significant reduction of absolute and relative ICC density [14]. Whether the intestinal dysmotility is a direct consequence of the abovementioned inflammatory peel has still to be evaluated. 

Fourth, a two-hit model, analogous to the pathomechanism of fetal spina bifida, was recently suggested. In addition to the congenital defect of the abdominal wall as the first hit, there seems to be a second hit: a progressive damage caused by mesenteric ischemia due to mechanical constriction of the prolapsed intestines through the abdominal defect, being too small for sufficient arterial perfusion [17]. Spontaneous intrauterine constrictive closure leads to incarceration and death of the prolapsed intestine and is associated with high premature birth and intrauterine fetal loss (vanishing gastroschisis). In this context, the intrauterine mortality of gastroschisis is currently not fully understood [18,19,20,21]. An incomplete closure of the defect may also lead to ischemia with loss of the incarcerated intestine (closing gastroschisis), but reports of this form are rare [22,23].

### 1.2. Prognosis

The prognosis of gastroschisis has improved due to advances and enhanced cooperation between the disciplines of maternal-fetal medicine, neonatal intensive care, and pediatric surgery. However, in the population of newborns with complex gastroschisis (gastroschisis additionally with at least one complication of the following four: intestinal atresia; perforation; necrotic segments; or volvulus) the mortality and morbidity rates are significantly higher compared to simple gastroschisis (gastroschisis without any of these complications). Significant differences in infants born with simple or complex gastroschisis also exist in clinical behavior, postoperative complications, and length of hospital stay. 

The cause of complex gastroschisis is not completely understood. Whether the reason lies in ongoing inflammation due to amniotic fluid exposure and peel formation, a complication of mesenteric constriction of the herniated intestines by the abdominal wall defect being too small, or in another hitherto unknown mechanism is not clear yet [24,25,26].

### 1.3. Fetal Treatment

It appears conclusive that, on the one hand, damage to the herniated intestines in gastroschisis results from amniotic fluid exposure and, on the other hand, the viscero-abdominal disproportion is prenatally acquired by extra abdominal growth of the herniated intestine. Complex gastroschisis itself also seems to be a consequence of the intrauterine pathology and not a complication acquired after birth. 

The purpose of prenatal treatment is to protect the prolapsed intestine from the amniotic fluid, which will reduce intestinal damage and allow the abdominal cavity to grow, with the best option being complete intestinal repositioning and closure of the abdomen. For these reasons, prenatal intervention to prevent the complications mentioned so far such as amniotic fluid infusion or exchange, gastroschisis repair or covering of the extra abdominal intestine seems to be a logical conclusion.

### 1.4. Aim of This Review

The aim of this review is therefore to evaluate the reported surgical approaches and techniques for the fetal treatment of gastroschisis with the focus on minimally invasive procedures.

## 2. Materials and Methods

### 2.1. Literature Research Strategy and Study Selection

A print and electronic literature search was completed from 1970 to February 2022 to identify published studies that were archived in PubMed, EmBase, and Medline. All available articles using the key words or medical subject headings (MeSH) “gastroschisis”, “fetoscopy”, “minimally invasive surgery”, “fetal surgery”, “amnioexchange”, “amnioinfusion” and “EXIT” published between 1970 and 2022 were included. Additional articles were identified by reviewing reference lists. After removing duplicates (*n* = 0), the titles and abstracts of 96 articles were reviewed initially and excluded based on predetermined criteria that included no fetal surgery or no relation to gastroschisis. 

A total of 52 articles passed this first selection process and were considered for full text review. From these articles, 10 were excluded for not identifying features of fetal surgery again. Figure 1 shows the selection process in detail.

### 2.2. Data Extraction and Quality Assessment

For potentially eligible publications full text articles were obtained and the quality and eligibility criteria applied independently. Each article was summarized, critiqued, and discussed by the entire working group to reduce bias and reach consensus. The following data were extracted from each article where available: “manuscript title; type of publication; type of (animal) model; surgical technique of creating gastroschisis; creation of gastroschisis on which day of pregnancy; fetuses with created gastroschisis; complications after creation of gastroschisis; number of fetuses that died during creation of gastroschisis; number of fetuses that died between creation of gastroschisis and fetal intervention; surgical approach to the fetus in the experimental group; intervention in the experimental group; number of experimental fetuses; notes/complications concerning the experimental group; intervention for gastroschisis on which day of pregnancy; time between creating gastroschisis and its intervention; time between intervention for gastroschisis and its evaluation; number of fetuses that died during fetal intervention for gastroschisis; number of fetuses that died between fetal intervention for gastroschisis and the end of gestation; percentage of fetuses that survived the intervention until term/evaluation; have healthy fetuses without gastroschisis been used as a healthy control group; number of healthy, unaffected controls; notes concerning the healthy group; have fetuses with gastroschisis without intervention been used as a control group; number of gastroschisis controls; notes concerning the control group; number of fetuses that died in the control group after creation of gastroschisis and the end of gestation/term/evaluation; day of pregnancy on which the experiment ended/terminated or the fetuses were delivered; notes”.

### 2.3. Statistical Analysis

No statistical or quantitative analysis was performed due to the heterogeneity and complexity of data for this descriptive review.

## 3. Results

A total of 42 reports were identified, including 30 experimental animal and 12 human studies (Figure 1, Table 1). 

Fetal interventions for gastroschisis in humans are only reported for EXIT procedures and amnio exchange. In animal models, particularly in the fetal sheep model, the various approaches to prenatal surgery can be divided into the repair or covering of gastroschisis by either open (using maternal laparotomy and hysterotomy) or minimally invasive fetal surgery (using fetoscopy). In addition, there are studies reported on EXIT procedures, prenatal amniotic infusion or amnio exchange, and open fetal esophageal ligation.

### 3.1. Open Fetal Repair

The first experimental report of open fetal repair of gastroschisis dates to 2003: A gastroschisis was created in the fetuses of six pregnant rabbits on day 27 of 31 days gestation by a 5 mm longitudinal paraumbilical incision in the left lower quadrant of the abdomen. The uterus was closed and then reopened two hours later for the open fetal repair of the previously created defect, which was completed by a 7-0 polypropylene running suture. All fetuses were retrieved five hours later and evaluated, which showed no difference of the serosal edema or wall thickening between the unrepaired and repaired fetuses [27].

Gastroschisis was surgically created also in two ovine fetuses on day 75. Both underwent open repair on day 100 of gestation by returning the eviscerated bowel into the abdomen and closing the abdominal wall primarily. With both fetuses surviving until delivery (day 135 of 145 gestational days) it could be demonstrated that fetal repair led to a grossly normal appearing bowel with few adhesions underlying a nearly scarless abdominal wall compared to unrepaired fetuses [28].

Another study reported three ovine fetuses with a simulated gastroschisis. An abdominal wall defect of 5 × 1 cm was surgically created on day 110–115 of 145 days gestation, which was repaired at the same time of its creation by closure and continuous suturing. Two of those three fetuses survived until spontaneous birth on day 145 and were euthanized and evaluated ten days later: the closure site was reported to be well-healed with only mild intra-abdominal adhesions [29].

Recently, three ovine fetuses with surgically created gastroschisis on day 76–80 of gestation were treated with open fetal surgery on day 99–101: The gastroschisis was created by a 1.5 cm horizontal incision in the left lower quadrant. For additional constriction of the eviscerated intestine, the abdominal opening was secured by a 1.5 cm diameter silicone rubber ring to prevent symmetric growth of the defect. Two of those three prenatally repaired fetuses survived until birth by cesarean section on day 139–141. One of those remaining two lambs survived until evaluation after 72 h of postnatal life with grossly normal appearing, normally functioning intestine, and minimal abdominal adhesions. The other one was euthanized 11 h after birth due to intestinal atresia and ileus [30]. 

### 3.2. Fetoscopic Repair

Kohl reported the fetoscopic creation of a gastroschisis in eight ovine fetuses on day 86 of 145. Six fetuses survived until day 110, where a fetoscopic repair was attempted by a simulated fetoscopic procedure: After maternal laparotomy and hysterotomy the fetuses were exteriorized, dried, and postured dorsoposterior on the maternal abdomen. Assessment of the fetoscopic treatment options was performed on the exteriorized fetuses using his fetoscopic setup and instrumentation. He was able to demonstrate that the repositioning of the eviscerated bowel was not possible due to the viscero-abdominal disproportion of the protruded intestine in relation to the small grown abdominal cavity. Furthermore, any attempt to return the herniated viscera into the fetal abdomen resulted in immediate and severe hemodynamic compromise of the fetoplacental circulation by stretching of the intra-abdominal umbilical arteries and vein, whereas a surgical enlargement of the defect could be performed without complications [31].

Our own group could reproduce the fetoscopic creation of a gastroschisis in eight fetuses on day 75 of 145, as well as prove the technical feasibility of a second fetoscopic procedure on day 105 for the evaluation of fetoscopic repair in the surviving seven fetuses: all fetoscopic attempts to reposition the intestine were unsuccessful—the defect, as well as the fetal abdominal cavity, were too small for the enlarged, rigidified, and peel-covered intestinal convolute [32].

### 3.3. Open Fetal Covering

Another option is the covering of prolapsed intestines to protect it from the amniotic fluid and restrain its extra-abdominal growth, therefore reducing the viscero-abdominal disproportion resulting in easier reduction and abdominal wall closure after birth without the use of a spring-loaded silo. Human data on this technique are lacking but results from several animal studies have been reported. 

Eight ovine fetuses underwent an open surgical coverage of the exteriorized bowel with an impermeable sheet of reinforced silastic on day 120. This patch was fashioned in a similar way to the postnatally applied silastic silo or Schuster patch. The gastroschisis was created by open surgery on day 80 and included a siliceous vascular loop with a 4 cm circumference that was placed around the bowel at the abdominal wall defect to provide gradual constriction during fetal growth. Of the eight animals treated with open fetal covering, six aborted after the fetal procedure. In the remaining two fetuses the villous atrophy and mesenteric venous and lymphatic dilatation were less pronounced compared to untreated controls [33,34].

In 2008, a collagen biomatrix was used in a sheep model for covering the gastroschisis: A gastroschisis was created by open fetal surgery in the left lower quadrant on day 75 of 145 gestation, and the intestines exposed to amniotic fluid in 14 fetuses. In nine of these fetuses, the intestine was manipulated back into the abdominal cavity, and the defect covered by a dual layer collagen biomatrix during the same intervention. The fetuses were retrieved 61 days later at the end of gestation and examined. Two fetuses died in the coverage group (22% mortality) for reasons unknown and one ewe had to be euthanized due to a soft tissue infection (33% overall fetal loss). The collagen biomatrix was replaced by scar tissue and integrated into the abdominal wall with a small herniation in one fetus [35]. Similar results were reported in 2013 by the same group: A total of 12 fetuses with surgically created gastroschisis were covered with a single or dual-layer porous scaffold from type I collagen derived from bovine Achilles tendon. Eight fetuses survived until delivery (33% mortality), in one fetus the scaffold had ruptured with a consecutive gastroschisis, in two lambs the bowel was partially extra-abdominal but still covered and in five animals, the abdomen was closed with the bowel situated intra-abdominally (success-rate 5/12, 42%) [36]. 

Another model for covering the gastroschisis was based on a fetal rat model described by Correia-Pinto in 2001 [37]. A gastroschisis was created in 36 fetuses on day 18.5 at mid-gestation: twelve fetuses were taken as controls without fetal intervention, in twelve fetuses the intestine was covered by a fibrin adhesive and in another twelve fetuses a cover with (P(NIPAAm-co-AAc) hydrogel, described by da Silva [38], was applied onto the fibrin adhesive. Another twelve fetuses without gastroschisis were kept as controls. The fetuses were delivered by cesarean section on day 31. The hydrogel appeared not to cause any damage to the serosa or to the villi of the bowel after removal, and no morphological changes occurred in the contact zone between the hydrogel and the bowel [39].

### 3.4. Fetoscopic Covering

Our group was able to demonstrate in two reports the technical feasibility of fetoscopically covering the prolapsed intestine with a natural latex bag. A gastroschisis was surgically created by two port fetoscopy (5mm camera and 3 mm instrument) at mid-gestation on day 75. After seven to twenty-eight days, a second fetoscopy was performed, covering the exteriorized intestines with a latex bag (sterilized condoms), which was fixed to the fetus with running or interrupted sutures by a percutaneous three port approach. We showed that the fetoscopic bag placement of the intestine led to a reduction in inflammatory damage, an increase in ICC density compared to untreated gastroschisis (in concordance to chicken egg models [13]), and a gradual reposition of the extra-abdominal bowel during gestation. In this way the viscero-abdominal disproportion could be reduced, although the fixation of the bags to the abdominal wall of the fetuses did not last until the end of gestation [14,40]. 

### 3.5. EXIT Procedures

An alternative to fetal intervention in utero is a surgical procedure immediately after delivery, when the fetus is still on the umbilical cord circulation (EXIT: Ex Utero Intrapartum Treatment Procedure). According to this procedure, the gastroschisis is treated with a primary closure of the prolapsed bowel before the newborn takes its first breath, which would result in bowel expansion [41]. To assess the feasibility of EXIT, prenatal ultrasound parameters are determined such as the Svetliza Reducibility Index (SRI), which is calculated by the largest diameter of the intestinal loops, the thickness of the loop wall, and the size of the abdominal wall defect [42].

In 2010, Zhang et al. were the first to describe this procedure treating two newborns with gastroschisis. In an average of 5.5 min, the EXIT was performed first, followed by interruption of the umbilical circulation. One of the newborns suffered from enteral torsion and feeding problems resulting in a lower weight than newborns of the same age but caught up by the fourth month [43]. 

In another study fetuses prenatally diagnosed with gastroschisis were treated with an EXIT-like procedure. After a planned cesarean section, in 5 min on average, the prolapsed intestine was completely repositioned into the fetal abdomen. The procedure used here differs from the original one in that no general anesthesia or medication for relaxation of the uterus were used. In one of five prospectively evaluated cases, the SRI was greater than 2.5 and thus not suitable for the EXIT-like procedure. In three fetuses, the intestine could be completely repositioned into the fetal abdomen, while in one case the EXIT-like procedure was aborted due to an iatrogenic injury, and a delayed closure using a customized Silo was carried out, such as in the six retrospectively evaluated control cases. The newborns in the EXIT-like group appeared to have a significantly better outcome with shorter mechanical ventilation, parenteral nutrition, and enteral feeding [41].

Another prospective and longitudinal study was designed with two groups including fetuses diagnosed with gastroschisis and excluding newborns with life-threatening malformations or intestinal perforation: while the experimental group was treated with EXIT (*n* = 23), the control group received a secondary closure with Silo placement (*n* = 15). Repositioning of the intestinal loops begins with the stomach followed by the intestine. If the umbilical cord circulation was still present after complete repositioning, the abdominal wall defect was sutured with 2-0 prolene. Otherwise, the umbilical cord was clamped first, and the wall closure was completed during resuscitation and ventilation. While two newborns from the EXIT group had to be treated with a Silo placement after all, one newborn was completely excluded from the study because a closure was only possible with delay (success-rate 20/23, 0.87%). At the end of the study, the experimental group included 20 and the control group 17 newborns. Mechanical ventilatory support was required in all cases with Silo placement (mean duration of 16 days); in the EXIT group in 38% of the cases (mean duration of 0.91 days) [44].

### 3.6. Amnioinfusion and Amnioexchange

This method is based on the idea that lowering the concentration of amnio toxins can reduce the damaging effects of amniotic fluid by infusing a volume of physiological saline into the amniotic cavity in fetuses with oligohydramnios (amnioinfusion), or by removing some of the amniotic fluid and replacing it with physiological saline in cases of fetuses with normal amniotic fluid volume (amnioexchange).

#### 3.6.1. Human Interventions

In 1996, two cases of fetuses with gastroschisis and oligohydramnios were treated with serial weekly amnioinfusions beginning from week 31 until delivery, which was due to spontaneous rupture of the membranes at 36 weeks. Both fetuses had minimal fibrous peel and the abdominal wall could be closed primarily [9].

Four years later, two fetuses with gastroschisis were treated with amnioinfusion (20G needle, warm saline) five times from week 27, the other once on week 31 due to oligohydramnios. The fetuses were retrieved by cesarean section in weeks 31 and 34, the bowel showed no signs of inflammatory peel, and the abdomen could be closed primarily in both [45]. 

A case series of 25 fetuses with gastroschisis receiving 1 to 5 amnioexchange procedures beginning from week 31 (range 26–38) and a mean duration between the first fetal intervention a birth of 32.5 days reported a change in arterial doppler velocimetry of the mesenteric superior artery. However, the postnatal macroscopic appearance of the fetal bowel failed to correlate with any Doppler index [46].

Furthermore, 30 human fetuses received a serial amnioexchange with a saline amnioinfusion. The first procedure was commenced at a mean gestational age of 32 weeks (range 26–38 weeks) and repeated up to six times (mean 2.5). Delivery of the fetuses occurred on a mean gestational age of 36.5 weeks (range 30–39.5 weeks) and therefore the median time of 4.5 weeks after starting serial amnioexchange [47].

Another eight human fetuses underwent serial amnioexchange with sterile saline solution from gestational week 30. Amniotic fluid samples taken at every procedure showed no significant differences in the concentration of the tested digestive enzymes (p-amylase and biliary salts) or inflammatory markers (C-reactive protein, serum amyloid A, Epidermal growth factor, Transforming growth factor b, interleukine-2 and 6, tumor necrosis factor alpha, myeloperoxidase) between the first and last amnioexchange procedure [48].

Five fetuses undergoing single (*n* = 3) or two times serial (*n* = 2) amnioexchange from week 32 were reported in 2004, all were delivered one to four weeks after commencing the procedure. A reduction of the fibrous coating was reported but no controls have been used [49].

One more case of serial amnioexchange starting on week 24 was repeated every four weeks until delivery in week 37. After birth, no inflammatory peel was found on the eviscerated intestines and the abdomen could be closed primarily [50].

A matched case control study on ten human fetuses with gastroschisis underwent an amnioexchange procedure from gestational week 32 on: four patients were amnioinfused twice, and two patients were amnioinfused three times. There were no significant differences found between the groups concerning ventilation; stay in NICU; time to oral nutrition; time to full oral feeding; total length of hospitalization; amount of perivisceritis; tension of closure or bowel atresia. Only curatization was found to be statistically longer in non-amnioinfused fetuses [51]. 

A prospective randomized and blinded trial of amnioexchange (*n* = 33, 18G needle, infusion with warm saline every two weeks from week 30 on until delivery) versus placebo (*n* = 39, sham intervention with subcutaneous injection, blinded mother) in human fetuses had to be stopped because the analysis failed to show any improvement in the amnioexchange arm and due to concern about potential side effects: the rate of intrauterine deaths was higher in the amnioexchange group [52]. 

#### 3.6.2. Animal Interventions

An ovine animal model with a hybrid technique of gastroschisis creation was reported, which included maternal laparotomy, uterine fixation of the amniotic membranes, and fetoscopic incision of the right lower abdominal wall on day 75 of pregnancy. Ten days later serial amnioinfusion was started and repeated every ten days on 12 fetuses, nine with untreated gastroschisis were taken as controls. A statistically significant increased thickness of serous fibrosis and plasma cell infiltration in the non-amnioinfused as compared with the amnioinfused fetuses could be demonstrated [11].

Continuous amnioinfusion by an indwelling catheter implanted on day 23 of 30 days gestation in seven rabbit fetuses failed to demonstrate a reduction of intestinal damage compared to operated fetuses without amnioinfusion (*n* = 8) or gastroschisis only (*n* = 7) [53]. 

A chicken embryo model was used for the study of amnioexchange in gastroschisis: In thirteen-day old chick egg embryos, a 2.5 mm defect was created with tweezers in the abdominal wall near the umbilical stalk and a catheter was placed into the amnio-allantoic cavity: Amnioexchange was performed with 0.075% saline solution once a day beginning the second day after gastroschisis creation for three consecutive days. Although the mortality of this model was up to 70%, it demonstrated that the mean of parasympathetic ganglia was measured 28% less in the gastroschisis-only compared to the healthy control group and that there was no significant difference between the control and gastroschisis-plus-exchange group. The intestinal contractility tests were better in the gastroschisis-plus-exchange compared to the gastroschisis-only group [54]. 

A similar chicken egg model was reported by Aktuğ: A gastroschisis-like defect in the umbilical stalk near the abdominal wall was made and intestinal loops were exteriorized on day 13 in chicken embryos. In the amnioexchange group (*n* = 47, 19 survivors), a catheter was inserted and an iso-osmolar infusion was instilled on days 15, 16, and 17. The embryos were retrieved on day 18 and compared to embryos without gastroschisis (*n* = 17, 10 survivors) or controls with gastroschisis but without amnioexchange (*n* = 37, 15 survivors). They demonstrated that the intestines of gastroschisis chickens without amnioexchange showed fibrous coating in addition to a severe inflammatory reaction, whereas the changes in the amnioexchange group consisted only of microscopic serosal edema, capillary proliferation, and an acute or chronic inflammatory reaction: no macroscopic fibrous coating was detected [55].

Similar to a study from 1995 [55], the same authors reported in a chicken egg model that the intestines of the gastroschisis embryos (*n* = 10) were thickened and covered by a fibrous peel, after amnioexchange (*n* = 10) there was no macroscopic fibrin deposition on the serosal surface. Microscopically, the intestinal muscular thickness was significantly decreased after amnioexchange compared to untreated gastroschisis. Similar results were obtained for bile acid levels. Aktuğ also described the protective effect of amnioexchange with physiologic saline solution compared to no fetal treatment and amnioexchange with dextrose: Each group of his chicken egg model consisted of ten embryos. The muscular and total mural thickness decreased after amnioexchange with saline compared to the other two groups [56].

Kanmaz et al. demonstrated in his chicken egg model that alkalization of amniotic fluid by amnioinfusion on day 15, 16, 17, and 18 with sodium bicarbonate (8.4% NaHCO3 solution (0.1 mL/100 g/d) led to a reduction in the signs of intestinal damage compared to amnioinfusion with a placebo (saline) [57].

Applying the model and techniques of Aktuğ and Kanmaz, Vargun was able to demonstrate in a chicken egg model, that amnioexchange prevented the decrease in ICC density encountered in damaged intestinal loops in gastroschisis [13,55,56,57,58].

An amnioinfusion every six hours after creation of a gastroschisis-like defect on day 25 until the delivery on day 32 showed in fetal rabbit models that the serosal thickness in the gastroschisis fetuses (*n* = 7) was significantly higher compared to healthy animals (*n* = 10) or animals treated with amnioinfusion (*n* = 7). No significant difference was found for intestinal wall thickness, muscular thickness, ganglion cells histology or infiltration of fibroblasts into the serosal layer [8].

An improvement of bowel morphology was observed after direct application of a nitric oxide donor (S-nitrosoglutathione, GSNO) onto the fetal eviscerated intestines. The NO donor was applied during open fetal surgery after the creation of the gastroschisis but can be instilled by amnioinfusion because of its liquid form [59].

One more chicken model yielded similar results of a normalized body weight and tibial length, near-normal intestinal wall thickness with slightly increased serosal width, near-normal intestinal DNA content and normal density of intramural ganglia after intrauterine dexamethasone application compared to embryos treated by saline amnioinfusion only [60].

A fetal sheep model of gastroschisis which was created by a hybrid approach (maternal laparotomy, manipulation of the uterus, and transuterine fetoscopy) was treated with amnioinfusion by an implantable port implanted during creation of the gastroschisis: a total of 46.7% of the fetuses survived the procedure, an amniotic infection occurred in 34.6% of cases. It could be demonstrated that continuous amnioinfusion appears technically feasible by implanted port devices [61].

Furthermore, the induction of fetal diuresis with intraamniotic furosemide injection led to a decrease of intestinal damage in a rat model but a comparison to amnioinfusion with saline only was not performed [62].

Transamniotic stem cell therapy (TRASCET) describes amnioinfusion with amniotic fluid mesenchymal stem cells (afMSCs) instead of normal saline. Feng et al. were able to demonstrate in a rabbit model that TRASCET, infused at the time of gastroschisis creation on day 23 of 33 gestation in New Zealand White rabbits, reduced the intestinal damage and segmental and total intestinal wall thicknesses compared to amnioinfusion with normal saline or no prenatal treatment [63,64].

Similar results were reported by Chalphin in a rat model [65,66]. 

### 3.7. Open Fetal Ligation of the Esophagus

Since amnioinfusion has shown to be of beneficial influence on reducing the intestinal damage associated with gastroschisis in an intrauterine sheep model, Lagausie et al. hypothesized that esophageal ligation will create a relative polyhydramnios, thereby leading to a dilution effect for the “amniotoxins” and reducing inflammation of the extruded bowel [51].

To test this hypothesis, the authors used their already existing model of gastroschisis in the fetal lamb combined with a left cervicotomy and ligation of the esophagus by open surgery and compared it with gastroschisis with or without amnioinfusion. Of 34 fetuses operated on in mid-gestation seven had gastroschisis and esophageal ligation. After the fetuses were killed at day 145, the extra abdominal bowel was processed for histologic examination (thickness of muscularis, thickness of serous fibrosis, and plasma cell infiltration). The results showed that, while amnioinfusion has a preventive effect on inflammation, ligation of the esophagus did not prevent digestive enzyme presence in the amniotic fluid [67].

## 4. Discussion

### 4.1. Postnatal Treatment of Gastroschisis

After birth, the unprotected intestine is exposed to dehydration, mechanical injuries, pressure necrosis, or infections [12]. Preventing these complications two postnatal tech-niques have become the standard for treatment of gastroschisis. This is completed by either repositioning the prolapsed intestine and subsequent closure of the abdominal wall (primary closure), or by temporarily covering the prolapsed intestine with a preformed silo bag that is fixed under the abdominal wall of the newborn and leads to a gradual reposition of the intestine. Delayed closure can then be carried out after a few days (sec-ondary closure) [68]. Secondary closure should be reserved for cases when primary closure is not possible. Modifications of the primary closure, techniques such as the sutureless repair of the abdominal defect, using the umbilical cord and adhesive coverings, have shown similar results with added benefits but an increased risk of umbilical hernia [69].

Aside from the increased risk of abdominal compartment syndrome after forced postnatal reposition, most newborns suffer from a prolonged intestinal transport disorder, which requires partial or complete parenteral nutrition for several weeks [70], leading to associated problems like hepatopathy, catheter infections, or sepsis [71,72]. Additional intestinal malformations such as volvulus, atresia, perforations, or necrosis also worsen the disease progression in cases of complex gastroschisis [24,73]. On the top, there is still an unexplained high rate of premature birth and intrauterine fetal demise in cases with gastroschisis, which cannot be reduced by waiting for postnatal management [18,74,75]. 

There has been a lot of controversy about the optimal surgical technique to be performed, as well as the best gestational age at delivery [76,77,78]. Some studies suggest that elective earlier delivery may reduce the late term mortality and general morbidity. However, a randomized trial of expectant management versus elective preterm delivery at 34 weeks did not demonstrate any benefit for the preterm born neonates [79].

### 4.2. The Idea of Prenatal Surgery for Gastroschisis

In recent years, advances in imaging techniques enabled early and detailed diagnosis of fetal anomalies in utero such as gastroschisis, which, if not treated on time, can lead to severe irreversible damage or even death. Data on the pathophysiology of gastroschisis demonstrated that factors contained in the amniotic fluid led to inflammatory and consecutive structural damage of the prolapsed intestine [33]. Prenatal interventions to protect the bowel from these compounds or to clean the amniotic fluid from them are just logical conclusions. These measures also have disadvantages: reduced inflammatory damage alone does not lead to a consecutive growth of the abdominal cavity, still resulting in an abdominal cavity too small to accommodate the bowel (viscero-abdominal disproportion), and a probable constricting ring leading to congestion or hypoperfusion of the prolapsed intestine. Whether the entity of complex gastroschisis can be prevented by cleaning the amniotic fluid is also still a matter of research. 

Furthermore, the aspect of wound healing should not be underestimated—regardless of the postnatal treatment method—it is only satisfactory for 50% of patients later in life. Wound healing that already begins intrauterine after successful fetal intervention may contribute to the outcome of the children [80].

With the above-mentioned factors in mind, the most effective prenatal management of gastroschisis should include the following criteria: First, protecting the bowel from the amniotic fluid; second, reducing the viscero-abdominal disproportion by either complete or partial reposition of the intestine into the abdominal cavity, or limiting the extra abdominal growth by covering it with a non-expandable sheet or bag. The increase of intra-abdominal pressure after covering the herniated intestine with a patch reduces the viscero-abdominal disproportion in relation to the still growing fetus. The covered defect will then be smaller and easier to manage after birth.

The results of previously reported techniques in either animal models or human applications are discussed in detail in terms of the outcome of fetuses with gastroschisis.

### 4.3. Open Fetal Repair

Several reports in sheep demonstrated the feasibility of open fetal surgery to reposition the intestine and close the abdominal wall, which led to a reduction of the intestinal damage compared to untreated controls. Unfortunately, the case number in those reports is small (2, 3 and 3 fetuses) and the mortality is high (up to 33%) [28,29,30]. Whether the mortality can be attributed to the procedure itself or the experimental pilot character of those three studies is hard to distinguish. As we know from open fetal surgery for myelomeningocele, the access to the fetus and closure of the uterus is not a relevant problem with a mortality of clearly less than 33% [81].

Nevertheless, repositioning the fetal intestine into an abdominal cavity being grown too small to include all the exteriorized and probably inflamed and swollen bowel, and thus risking an intrauterine fetal abdominal compartment syndrome, will not be an option for prenatal intervention. 

### 4.4. Fetoscopic Repair

Open fetal surgery carries the risk of premature rupture of membranes and the need for cesarean section in all further pregnancies due to the hysterotomy. 

Minimally invasive procedures might reduce the trauma to the uterus and amniotic membranes. A hybrid laparotomy—fetoscopic approach was developed by Belfort et al. which led to a significant decrease of chorioamniotic separation and premature rupture of membranes compared to open and complete fetoscopic procedures [82]. Minimally invasive repair of gastroschisis has been evaluated by Kohl et al., although their technique included the extraction of the fetus and its placement onto the abdomen of the ewe with assessment using fetoscopic instrumentation [31]. The first true percutaneous fetoscopic intervention for gastroschisis repair in sheep was described by us in 2012. Even when applying a three-port approach similar to laparoscopic surgery, fetoscopic repositioning of the eviscerated bowel, which was enlarged and rigidified due to 21 days of amniotic fluid exposure, was not possible [32]. Enlargement of the defect was reported as being possible by Kohl et al. but has not been tried fetoscopically. Enlargement of the defect by incising the abdominal wall via a laparotomy will also not enlarge the volume of the abdominal cavity and carries the risk of damaging umbilical arteries and veins [31,32].

### 4.5. Open Fetal Covering

The covering of gastroschisis by open fetal surgery led to a reduction of inflammatory peel formation, intestinal damage, as well as an improvement in the Cajal pacemaker cell density in several animal models [28,34,35,36]. Additional to the abdominal wall defect, the group of Langer used a silastic vessel loop acting as a constrictor for the growing fetus to simulate hypoperfusion. Interestingly they found a relevant negative effect of amniotic fluid exposure on bowel contractility that was entirely independent of bowel constriction and attributed to the inflammatory damage by the amniotic fluid. The results of Roelofs et al. have to be seen with caution: the surgical intervention of covering the gastroschisis was performed at the same time of its creation (day 79 of 140 days gestation). Therefore, fetal surgery was performed on a naive intestine instead of bowel that had a long duration of exposure to the amniotic fluid and therefore inflammatory changes. This model is thus not suitable to evaluate whether fetal surgery might alleviate and decrease a pre-existing damage to the prolapsed intestines [35,36]. 

### 4.6. Fetoscopic Covering

The encouraging results of covering the gastroschisis with an impermeable membrane by open fetal surgery seven to 28 days of its creation could be repeated by applying a completely fetoscopic approach and bag placement of the intestine [14,40]. As the bags dislocated partially or completely during late gestation, refinement and technical improvement of the instruments and methods are needed [83]. In contrast to our technique of amnio distension by amnioinfusion and therefore working in an under-water environment, other authors suggest the use of partial amniotic carbon dioxide insufflation (PACI) for amnio distension and thus much better instrumentation of the fetus. The consequences of both techniques onto the uterus and chorioamniotic membranes have to be directly compared in animal studies [84,85].

### 4.7. EXIT Procedures

Complete reposition of gastroschisis as soon as possible in the least traumatic way may avoid complications [12,86].

To assess the feasibility of EXIT and similar procedures, the SRI index used so far seems to be indicative, as a smaller abdominal wall defect makes repositioning of the bowel loops more difficult, especially under time pressure. In these cases, surgical enlargement of the defect would be a solution but is unfeasible without anesthesia [42].

The fact that Cisneros et al. designed their study excluding complex gastroschisis shows that this procedure, especially without anesthesia, is applicable only in selected cases with simple gastroschisis [44]. 

The reported differences in outcomes between the experimental and control groups are not always based on significant results due to the small number of case studies; in particular, a critical view should be taken of the shorter hospital stay. It is even questionable whether the better outcome concerning the time of nutritional transition and hospital discharge is due to surgical technique or preterm birth in the EXIT-like group compared with term birth in the Silo group [41]. 

Based on these results, the EXIT method could have a better outcome for simple gastroschisis, but these should be confirmed by a greater number of cases, as mentioned by the authors. In any case, it does not seem to be a solution for complex gastroschisis.

### 4.8. Amnioinfusion and Amnioexchange

Evaluating amnioinfusion or amnioexchange in animal models from the 1990s on, several groups were able to demonstrate that amnioexchange removes digestive substances in the amniotic fluid and consecutively reduces intestinal inflammation and bowel wall damage in chicken egg [8,54,55,56] or ovine models [11] or human fetuses [45,46,47,48,49,51,52]. This effort culminated into a prospective randomized controlled multicenter trial of amnioexchange for fetal gastroschisis in 2019 [52] which had to be stopped as interim analysis failed to show any benefit of amnioexchange and demonstrated a potential excess risk of fetal death in the amnioexchange group. Although one may argue that the amnioexchange procedure (started at week 30 and repeated every two weeks until delivery with an infusion volume of 300 mL) resulted in less and therefore ineffective dilution of amniotic fluids and that having only a transient effect, applying a twice weekly interval may be too long for effective dilution. Nevertheless, more frequent amnioinfusion by puncture with a 20-gauge needle increases procedural morbidity and mortality as demonstrated in this study. An improvement may be the implantation of a venous port-like system with a tube inserted into the amniotic cavity which is connected to a subcutaneously inserted port chamber. The resulting long-term amnioinfusion reduces the trauma to the uterine cavity to just the implantation of the port system [87,88]. Whether this method may have a protective effect on the bowel of gastroschisis with concomitantly reduced procedural morbidity and mortality, must be evaluated. Furthermore, the optimal medium for amnioinfusion (saline, TRASCET or alkalization) has to be determined. 

### 4.9. Open Fetal Ligation of the Esophagus

As mentioned previously serial amnioinfusion still has a significant procedural morbidity and mortality due to recurrent trauma to the uterine cavity. On these grounds esophageal ligation was a promising hypothesized one-time intervention in order to create polyhydramnios and thereby prevent inflammation of the bowel by a dilution effect. The results of a single animal study showed that ligation of the esophagus did not prevent digestive enzyme presence in the amniotic fluid, but actually increased intestinal damage. Lagausie et al. might have prevented the intestinal damage by ligation of the rectum or anus instead of the esophagus. Correia-Pinto et al. have ligated the anus in a rat gastroschisis model and showed that cessation of meconium discharge can prevent damage of extruded bowels [89]. Ligation of the fetal esophagus or rectum will also lead to post-partum obstruction or chronic stenosis, which will eventually result in much more additional morbidity.

In conclusion, esophageal ligature does not appear to be a feasible fetal therapy of gastroschisis.

### 4.10. Limitations

Despite the probable benefits of fetal surgery for gastroschisis, the reported techniques all have their own relevant limitations. Additionally, this review includes experimental animal models, cases, or case series of human procedures and one prospective randomized blinded trial. The heterogeneity of the deducted data is therefore high. 

Any conclusions have to be discussed only in the context of the specific animal model or human procedure, especially focusing on the technical details and the grade of “human simulation” of the animal models used. 

#### 4.10.1. Limitations of the Animal Models

The study by Till et al. described a rabbit model of fetal surgical repair of gastroschisis, but the exposure of the prolapsed intestine to the amniotic fluid lasted only two hours (of 32 days of gestation). It is questionable, whether an exposition to amniotic fluid for just two hours will sufficiently simulate human gastroschisis with months long exposition of the intestine, and its impact on intestinal inflammation, peel formation and damage. Any conclusions from those studies concerning the effectiveness of the surgical procedure, and the feasibility of intestinal reposition followed by abdominal closure of the just freshly exteriorized, and thus not damaged, intestine will have to be drawn with caution. Some other reported animal models also do not simulate human gastroschisis or the proposed fetal intervention well [27,31,35]. Therefore, there is a need to define the criteria for animal models which are used to simulate human gastroschisis:(I)The abdominal wall defect has to occur as early as possible in gestation. Whether by open or fetoscopic surgical creation or by toxic/nutritional induction is less relevant;(II)An early abdominal wall defect will lead to a relevant period of time of contact of the protruded intestines with the amniotic fluid. As we were able to demonstrate a correlation of amniotic fluid exposure and intestinal damage, only the prolonged amniotic fluid exposure will lead to intestinal inflammatory damage and simulate the pathology of gastroschisis [90]. Thus, any report of fetal surgery for gastroschisis right after its creation will be heavily confounded [27,31];(III)Furthermore, a relevant period of time after the fetal intervention has to pass before the fetuses are delivered and examined. This will ensure that the therapeutic mechanism of the fetal procedure has enough time to take effect as compared to untreated fetuses with ongoing intestinal damage. Only this setup will ensure that the fetal intervention reverses the pre-existing damage of the intestines;(IV)This leads to the prerequisite that every study should include healthy (no gastroschisis) and control groups (untreated gastroschisis) to compare the effect of fetal surgery to the naive and diseased intestine.

#### 4.10.2. Technical Limitations of Fetal Surgery for Gastroschisis

One of the complications associated with fetal surgery is preterm, prelabor rupture of membranes (PPROM) due to chorioamniotic membrane separation (CMS). The fluid-filled chorionic cavity separates the chorionic and amniotic membranes in the first trimester. As pregnancy progresses, the amniotic cavity expands, resulting in a vanishing chorionic cavity due to the fusion of the chorionic and amniotic membranes. In case of CMS the amnion lifts partially or completely off the chorion, usually seen after traumatic entry into the amniotic cavity, and is complicated by the impaired wound healing of the membrane injury from poor vascularization and lack of innervation [91]. Therefore, the risk for miscarriage, in utero fetal death, umbilical cord complications, preterm delivery, and amniotic band formation is increased.

Especially minimally invasive fetoscopic procedures seem to have a high risk for PRROM after CMS: puncturing the membranes with the fetoscopic port exploits the stretchability of the membranes [92]. A solution for this problem which limits the benefits of fetoscopy could be the Seldinger technique as a way of minimizing local membrane trauma and preventing chorioamniotic separation during fetoscopic port insertion [93,94]. More recently, suturing the fetal membranes to the uterine wall before port insertion to prevent chorioamniotic separation has been reported [83]. Larger cohort studies are still required to confirm these and alternative port insertion techniques, considering that the structure and integrity of human fetal membranes are different from animal models. 

#### 4.10.3. Ethical Limitations of Fetal Surgery for Gastroschisis

There are ethical limitations concerning the use of animal models for the prenatal management of gastroschisis. As the gastroschisis has been created by surgery, which is a burden on the animal and fetus even before the actual experiment, an animal model of spontaneous complex gastroschisis or its induction by mutagenic substances might be much less strain on the animals and thus refine (“3R”) the animal model. To the best of our knowledge, no large animal model exists, which induces gastroschisis by other means than fetal surgery. Reports can be found on gastroschisis induction by retinoic acid, but not in the sheep model, which is essential for examining the effect of surgical interventions due to its size [95].

In our opinion, it is absolutely clear that animal models should only be the ultimate and last step before human application of a new surgical technique for fetal surgery of gastroschisis. However, as the effect of any prenatal therapy can only be evaluated in a growing fetus and its mother, experiments in animal models appear mandatory. In order to affect the “3Rs”: Reduce the number of animals, to refine the surgical technique before experimenting in animals, and to replace animals by alternative models, we suggest a step-up approach of evaluating any new technique or instrument for fetal surgery of gastroschisis: First, evaluation of the techniques and instruments in inanimate models (box trainers, puppets, custom-made simulators of uterine surgery) with revision according to the experimental findings. Second: after successfully passing step 1, subsequent evaluation of the revised techniques and instruments in euthanized animals that result from other studies, for example, heart valve replacements. Thus, the sacrificed animals will serve another additional scientific purpose which adds up to a reduction in the use of animals: Further refinement of the instruments and techniques according to the results from the cadaver lab studies. Third and last: examination of the instruments and techniques in live animal fetal models after successfully passing steps 1 and 2.

Fetal surgery for a relative benign malformation such as gastroschisis certainly is very controversial. Although the morbidity in the neonatal period may be significant, the overall survival rate is more than 90% and with good long-term results. This must be weighed against the risk of intrauterine death or prematurity in addition to possible surgical complication to the intrauterine repair of the gastroschisis [96]. Interestingly, the entity of complex gastroschisis carries a much higher morbidity and mortality than the overall population of children affected with gastroschisis and much more strikingly when compared to the entity of simple gastroschisis [24,96]. Therefore, complex gastroschisis should be the primary target for fetal intervention, as patients with simple gastroschisis have no significant impairment in gastrointestinal function or Health-Related Quality of Life compared with healthy controls in adolescence and adulthood [81]. For these reasons, and in view of the complications of fetal interventions mentioned so far, which pose risks to both the mother and the fetus [97,98], simple gastroschisis does not justify fetal intervention.

Complicating this issue is the circumstance, that there are currently no hard data which may help to reliably identify complex gastroschisis and differentiate it from simple cases by ultrasound or serological markers long enough prenatally to intervene in time before the damage to the intestine occurs. It appears that prenatal bowel dilation may be indicative for complex gastroschisis, but the sensitivity has been reported in some cases as low as 50% [99]. Thus, even if a minimally invasive fetal surgical intervention for gastroschisis with acceptable maternal and fetal morbidity and mortality has been developed, the prenatal selection of the fetuses which benefit from surgery is currently not possible. Further research of ultrasound and serological markers may result in specific prenatal diagnoses in the near future [100,101].

Prenatal diagnostics itself is already controversial from an ethical point of view: it is possible to detect diseases that can affect the fetus already at the beginning of pregnancy. As long as there are no safe measures that can be taken intrauterine and cure the disease early, many parents might then draw the consequence to terminate their pregnancy.

Another reason why fetal surgery needs to be well considered is that after open fetal surgery, not only the current pregnancy but also all future pregnancies must be delivered by cesarean section. Whereas, and that is why we emphasize the importance of fetoscopy, with minimally invasive fetal surgery, vaginal delivery may be considered for the current and future gestations [83].

## 5. Conclusions

The most promising techniques of prenatal management of complex gastroschisis appear to be amnioexchange and fetoscopic covering of the protruded intestine [24]. 

In contrast to animal studies, current level 1 data on amnioexchange in humans are not encouraging due to ineffective dilution and procedural related mortality. The advent of continuous amnioinfusion through a subcutaneously implanted port system may be a relevant improvement of this technique [88,89]. However, even successful amnioexchange will not reduce the viscero-abdominal disproportion which complicates the repositioning of the intestines into the abdominal cavity. Bimanual fetoscopic surgery might be an option: experimental data demonstrate that covering the prolapsed intestines with an impermeable membrane not only reduces the intestinal damage but also improves growth of the abdominal cavity [14,28,30,32,35,102].

In our opinion, a fetoscopic minimally invasive approach to the fetus and covering the gastroschisis appears to be the best surgical option: although those promising results have been reported in animal models and humans, further studies need to evaluate the optimal materials, instruments, and techniques of fetoscopic surgery as a potential management especially for cases of complex gastroschisis [24]. Therefore, future research should be directed to establish an animal model of complex gastroschisis which is large enough to be undergoing surgical interventions, preferably by minimally invasive procedures.

After establishment of the proper prenatal diagnosis of complex gastroschisis and patient selection as well as the surgical technique, we would propose the algorithm shown in Figure 2 for the fetal management of gastroschisis: 

## Figures and Tables

**Figure 1 children-09-00416-f001:**
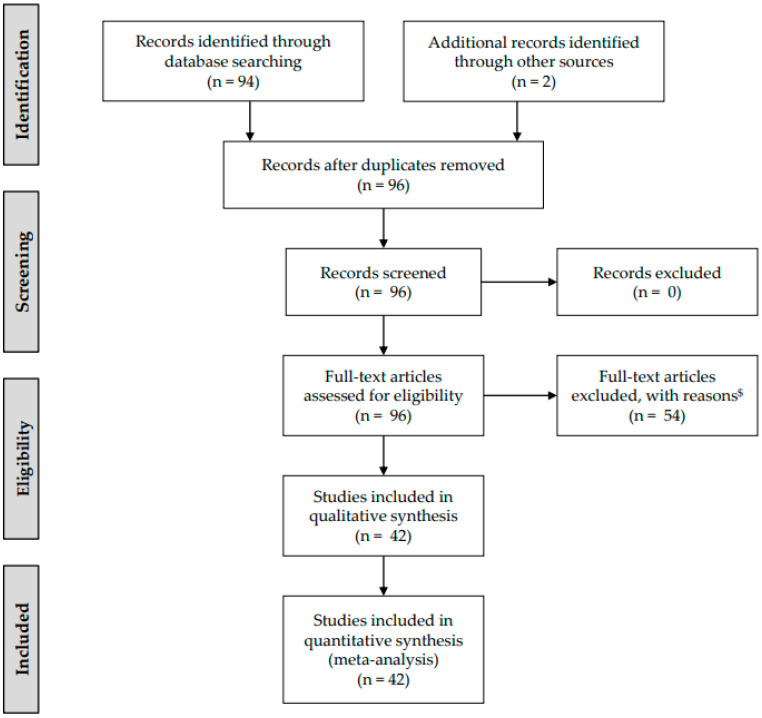
The PRISMA flow chart of evaluation and critical inclusion of the reviewed reports. $: not reporting on fetal surgery.

**Figure 2 children-09-00416-f002:**
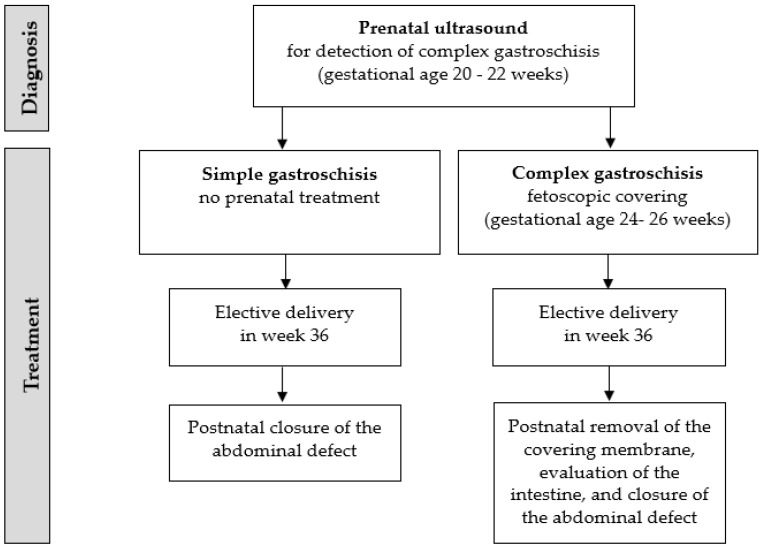
The algorithm for fetal management of gastroschisis.

**Table 1 children-09-00416-t001:** The included reports on fetal surgery for gastroschisis.

Author	Year	Study	Creation	Day	Approach	Intervention	Exposition	Treatment	Healthy	Control	End
Dommergues	1996	human (CS)	-	-	AI10	30 + 5 and 32 weeks	-	4 and 5 weeks	no	no	36 and 36.5 weeks
Luton	1999	human (CCS)	-	-	AI12	32 + 3 (26–38) weeks	-	4.6 (0.5–10) weeks	no	yes	35–38 weeks
Sapin	2000	human (CS)	-	-	A14	27 weeks	-	4 and 3 weeks	no	no	31 and 34 weeks
Volumenie	2001	human (CS)	-	-	AI13	31 (26–38) weeks	-	32 days	no	no	35.5 weeks
Burc	2004	human (CS)	-	-	A13	32 (26–38) weeks	-	4.5 weeks	no	no	36.5 weeks
Turkota	2004	human (CS)	-	-	A12	32 weeks in 4, 36 in one	-	4 weeks	no	no	36 weeks
Midrio	2007	human (CS)	-	-	AI10	30 weeks	-	5 weeks	no	no	35 weeks
Zhi-tao Zhang	2010	human (CS)	-	-	EXIT	N/N	-	-	no	no	-
Demir	2013	human (CR)	-	-	AI11	24 weeks	-	13 weeks	no	no	37 weeks
Cisneros-Gasca	2014	human (CS)	-	-	EXIT	34–40 weeks	-	-	no	yes	-
Oliveira	2017	human (CS)	-	-	EXIT	34–37 weeks	-	-	no	yes	-
Luton	2019	human (PRT)	-	-	AI10	30 weeks	-	5 weeks	no	yes	35 weeks
Langer	1990	fetal sheep *	O1	80	O7	120	40	15	yes	yes	135
Luton	2000	fetal sheep *	H1	75	AI1	85	10	60	yes	yes	145
Guys	2002	fetal sheep *	H2	80	H1	80	0	57	yes	no	137
Lagausie	2002	fetal sheep *	H1	75	O2	70–80	0	65–75	yes	yes	145
Roelofs	2008	fetal sheep *	O4	79	O3	79	0	61	no	yes	140
Kohl	2009	fetal sheep *	F1	86	O4	110	23.5	0	no	no	110
Stephenson	2010	fetal sheep *	H1	75	O5	100	25	35	yes	yes	135
Sun	2011	fetal sheep *	O2	112.5	O5	112	0	32	no	no	145
Bergholz	2012	fetal sheep *	F1	75	F1	105	30	27	no	no	132
Roelofs	2013	fetal sheep *	O4	79	O3	79	0	61	no	no	140
Krebs	2014	fetal sheep *	F1	77	F2	99	22	36	yes	yes	135
Anderson	2019	fetal sheep *	O2	78	O5	100	22	18	no	yes	140
Bergholz	2020	fetal sheep *	F1	75	F2	96	19	47	no	no	143
Till	2003	fetal rabbit *	O4	27	O5	27	0	0	yes	yes	27
Ashrafi	2008	fetal rabbit *	O3	25	AI3	25	6h	7	yes	yes	32
Feng	2016	fetal rat *	O4	18	AI4	18	0	3	no	yes	21.5
Feng	2017	fetal rabbit *	O4	23	AI4	23	0	10	no	yes	33
Goncalves	2010	fetal rat *	O3	18.5	O6	18.5	0	3	yes	yes	21.5
Hakguder	2011	fetal rat *	O3	18.5	AI5	20	1.5	1.5	yes	yes	21.5
Gonçalves	2015	fetal rat *	O3	18.5	AI6	18.5	0	3	yes	yes	21.5
Chalphin	2020	fetal rat *	O3	18	AI4	18	0	4	no	yes	22
Chalphin	2020	fetal rat *	O3	18	AI4	18	0	4	no	yes	22
Aktug	1995	chick embryo *	O5	13	AI7	15, 16 and 17	2	3	yes	yes	18
Aktug	1998	chick embryo *	O5	13	AI7	15, 16 and 17	2	3	yes	yes	18
Aktug	1998	chick embryo *	O5	13	AI7	15, 16 and 17	2	3	no	yes	18
Kanmaz	2001	chick embryo *	O5	13	AI7	15, 16, 17 and 18	2	3	yes	yes	18
Sencan	2002	chick embryo *	O5	13	AI7	15, 16 and 17	2	3	yes	yes	18
Yu	2004	chick embryo *	O5	15	AI8	17	2	2	yes	yes	19
Vargun	2007	chick embryo *	O5	13	AI9	15, 16 and 17	2	5	yes	yes	18
Munoz	2002	fetal rabbit *	O3	23	AI7	daily	0	7	yes	yes	30

* = experimental animal study; PRT = prospective randomized trial; CS = case series; CR = case report; CCS = case control study; for creation of the gastroschisis: O1 = open fetal surgery and incision of the right lower abdomen, placement of a 4 cm circumference vessel loop as constrictor; O2 = open fetal surgery and incision of the abdomen, placement of a 1.5 cm diameter silicone ring in the defect; O3 = open fetal surgery and incision at the right of the fetal umbilicus; O4 = ex 10: open fetal surgery and an incision was made in the left lower or right lower quadrant; O5 = chick egg model (after perforating amnio allantoic membrane, a 2.5-mm defect was created with tweezers on abdominal wall near the umbilical stalk); H1 = maternal laparotomy, uterine purse string, fetoscopic incision in the right lower quadrant; H2 = ex 14: maternal laparotomy and uterine fetoscopy: fetal abdominal wall incision and exteriorization of omentum and intestine; F1 = percutaneous fetoscopic incision in the left lower quadrant; for the approach to the fetus and management of gastroschisis: O1 = open fetal surgery; O2 = open fetal surgery: creation of gastroschisis and esophageal ligature simultaneously (gastroschisis only controls were created by hybrid fetoscopy); O3 = open fetal surgery: covering the prolapsed intestine with collagen scaffold; O4 = open fetal surgery: exteriorization of the fetus outside of the uterus and maternal abdomen: assessing fetoscopic options on the exteriorized fetus, fetoscopic repositioning and fetoscopic closure of the defect; O5 = open fetal surgery: repositioning of the intestine and closure of the defect; O6 = open fetal surgery: covering the prolapsed intestine with fibrin adhesive and dry hydrogel; O7 = open fetal surgery; covering the prolapsed intestine with reinforced silastic; H1 = hybrid procedure: maternal laparotomy, uterine manipulation, fetoscopy, implantation of a catheter or port for continuous amnioinfusion; F1 = ex 2: percutaneous fetoscopic surgery: repositioning of the intestine and closure of the defect; F2 = percutaneous fetoscopic surgery: covering the prolapsed intestine with a bag; AI1 = ex 5: ultrasound guided amnioinfusion every 10 days until term—amniocentesis by 20 G needle; AI2 = ex 6: ultrasound guided amniocentesis by 18 G needle; AI3 = open fetal surgery: implantation of an arrow CVC during creation of the gastroschisis, consecutive amnioinfusion; AI4 = open fetal surgery: amnioinfusion with TRASCET during creation of the gastroschisis; AI5 = amnioinfusion: intraamniotic furosemide injection; AI6 = open fetal surgery: amnioinfusion during creation of the gastroschisis by applying 30 μL of S-nitrosoglutathion to the bowel in three varying doses; AI7 = open fetal surgery: serial amnioinfusion with a catheter that was placed into the amnio-allantoic cavity during creation of the gastroschisis and AF exchange was performed with 0.075% saline solution; AI8 = amnioinfusion: intra-amniotic dexamethasone injection; AI9 = AI7 plus one group of amnioinfusion with bicarbonate; AI10 = ultrasound guided amnio exchange every 2 weeks until term—amniocentesis by 18 G needle; AI11 = ultrasound guided amnio exchange every 4 weeks until term; AI12 = ultrasound guided amnio exchange every 2 weeks, maximum two to three times; AI13 = ultrasound guided amnio exchange every 2–3 weeks—amniocentesis by 20 G needle; AI14 = ultrasound guided amnioinfusion once and every week; EXIT = EXIT procedure.

## Data Availability

Not applicable.

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
