# Peer review of "Fetal Surgery for Gastroschisis—A Review with Emphasis on Minimally Invasive Procedures"

_children, 2022, doi:10.3390/children9030416_

Round 1
Reviewer 1 Report
The subject is relevant, and the paper is well-written and structured although there are several repetitions. Fetal surgery for a relative benign malformation as gastroschisis certainly is very controversial. Although the morbidity in the neonatal period may be significant, the survival rate I more than 90% and with good long-term results. This must be weighed against the risk of intrauterine death or prematurity in addition to possible surgical complication to the intrauterine repair of the gastroschisis. These ethical considerations must be included in the manuscript.
I do agree with the authors that eventual uterine repair should be reserved the cases with complex gastroschisis, but who can they we diagnosed with certainty? And what GA would the authors recommend a repair? This has also to be weighed against the possibility of a termination as decided by the parents which has become rather common in the last decade in European communities.
Doe the authors have any thoughts of the likelihood for an animal model with “spontaneous” gastroschisis comparable to the human characteristics for gastroschisis? Considering the 3R´s in animal experimental studies will the authors the suggest stop doing animal studies on this subject?
All these ethical issues should be commented and contained in a paragraph on ethical considerations.
Long-trem outcome: Snoep MC et al. Early Human Dev 2020;141:104936
Survial: Brebner A et al. J Matern Fetal Neonatal Med 2020;33:1725-31
Author Response
Reviewer 1
The subject is relevant, and the paper is well-written and structured although there are several repetitions.
Thank you very much, we are very grateful for your expert opinion. We have taken out repetitions in the text to the best of our ability for a better reading flow and accuracy.
Fetal surgery for a relative benign malformation as gastroschisis certainly is very controversial. Although the morbidity in the neonatal period may be significant, the survival rate is more than 90% and with good long-term results. This must be weighed against the risk of intrauterine death or prematurity in addition to possible surgical complication to the intrauterine repair of the gastroschisis. These ethical considerations must be included in the manuscript.
Thank you very much for this important point we clearly left out before: Please find under "Ethical limitations of fetal surgery for gastroschisis" where we have further elaborated the ethical aspects according to your comments. Additionally, we described in our manuscript the entity of complex gastroschisis, with its significantly increased morbidity and mortality compared to the simple form of gastroschisis, as the main probable target for any fetal intervention. Therefore, fetuses with simple gastroschisis, which have an excellent postnatal survival, (although it can not be diagnosed prenatally yet) should definitely be excluded from any form of fetal intervention. The proposed algorithm in our revised conclusion explicitly excludes those cases.
The “Ethical limitations of fetal surgery for gastroschisis” paragraph reads as follows:
“ Fetal surgery for a relative benign malformation such as gastroschisis certainly is very controversial. Although the morbidity in the neonatal period may be significant, the overall survival rate is more than 90% and with good long-term results. This must be weighed against the risk of intrauterine death or prematurity in addition to possible surgical complication to the intrauterine repair of the gastroschisis [105]. Interestingly, the entitiy of complex gastroschisis carries a much higher morbidity and mortality than the overall population of children affected with gastrochisis and much more strikingly when compared to the entity of simple gastroschisis [24,105]. Therefore, complex gastroschisis should be the primary target for fetal intervention, as patients with simple gastroschisis have no significant impairment in gastrointestinal function or Health-Related Quality of Life compared with healthy controls in adolescence and adulthood [104]. For these reasons, and in view of the complications of fetal interventions mentioned so far, which pose risks to both the mother and the fetus [98,99], simple gastroschisis does not justify fetal intervention.
Complicating this issue is the circumstance, that there are currently no hard data which may help to reliably identify complex gastroschisis and differentiate it from simple cases by ultrasound or serological markers long enough prenatally to intervene in time before the damage to the intestine occurs. It appears that prenatal bowel dilation may be indicative for complex gastroschisis, but the sensitivity has been reported in some cases as low as 50% [100]. Thus, even if a minimally invasive fetal surgical intervention for gastroschisis with acceptable maternal and fetal morbidity and mortality has been developed, the prenatal selection of the fetuses which benefit from surgery is currently not possible. Further research of ultrasound and serological markers may result in specific prenatal diagnoses in the near future [101,102].
Prenatal diagnostics itself is already controversial from an ethical point of view: It is possible to detect diseases that can affect the fetus already at the beginning of pregnancy. But as long as there are no safe measures that can be taken intrauterine and cure the disease early, many parents might draw the consequence to terminate their pregnancy.
Another reason why fetal surgery needs to be well considered is that after open fetal surgery, not only the current pregnancy but also all future pregnancies must be delivered by cesarean section. Whereas and that is why we emphasize on fetoscopy, with minimally invasive fetal surgery, vaginal delivery may be considered for the current and future gestations [97].“
I do agree with the authors that eventual uterine repair should be reserved for cases with complex gastroschisis, but who can they be diagnosed with certainty?
We completely agree with you: Currently, there is no reliable ultrasound or serological marker to prenatally diagnose complex gastroschisis with certainty. Some measurements of bowel diameters during the 20-22 week of gestation appear to be promising, but hard data is not available [Joyeux, Complex gastroschisis: a new indication for fetal surgery? doi: 10.1002/uog.24759]. But we are confident that in the near future the prenatal identification of complex gastroschisis will be possible early enough to open the door for a probable fetal intervention of those selected cases.
And what GA would the authors recommend a repair? This has also to be weighed against the possibility of a termination as decided by the parents which has become rather common in the last decade in European communities.
Indeed, we see a rise in termination of otherwise healthy children with „only“ having a gastroschisis. Please see our revised discussion paragraph where we suggest an algorithm with fetal intervention during week 24-26, which will give the intestine enough time to heal before birth and also deliver fetuses not too premature if any complication after the fetal intervention occurs, such as premature rupture of membranes.
Do the authors have any thoughts of the likelihood for an animal model with “spontaneous” gastroschisis comparable to the human characteristics for gastroschisis? Considering the 3R´s in animal experimental studies will the authors suggest stop doing animal studies on this subject?
Thank you very much for this comment: As the gastroschisis has been created by surgery, which is a burden on the animal and fetus even before the actual experiment, an animal model of spontaneous complex gastroschisis or its induction by mutagenic substances might be much less strain on the animals and thus refine („3R“) the animal model. But to the best of our knowledge, no large animal model exists, which induces gastroschisis by other means than fetal surgery. Reports can be found on gastroschisis induction by retinoic acid, but not in the sheep model, which is essential for examining the effect of surgical interventions due to its size.[Paulo Roberto Veiga Quemelo 1 , Charles Marques Lourenço, Luiz Cesar Peres. Teratogenic effect of retinoic acid in swiss mice. Acta Cir Bras. Nov-Dec 2007;22(6):451-6.PMID: 18235933]
In our opinion, it is absolutely clear that animal models should only be the ultimate and last step before human application of a new surgical technique for fetal surgery of gastroschisis. But as the effect of any prenatal therapy can only be evaluated in a growing fetus and its mother, experiments in animal models appear mandatory. In order to „3R“: Reduce the number of animals, to refine the surgical technique before experimenting in animals and to replace animals by alternative models, we suggest a step up approach of evaluating any new technique or instrument for fetal surgery of gastroschisis: First - evaluation of the techniques and instruments in inanimate models (box trainers, puppets, custom made simulators of uterine surgery) with revision according to the experimental findings. Second: After successfully passing step 1 - evaluation of the revised techniques and instruments in euthanized animals that result from other studies, for example heart valve replacements. Thus, the sacrificed animals will serve another additional scientific purpose which adds up to the reduction of animals : Further refinement of the instruments and techniques according to the results from the cadaver lab studies. Third and last: Examination of the instruments and techniques in live animal fetal models after successfully passing step 1 and 2.
All these ethical issues should be commented on and contained in a paragraph on ethical considerations.
Long-term outcome: Snoep MC et al. Early Human Dev 2020;141:104936
Survival: Brebner A et al. J Matern Fetal Neonatal Med 2020;33:1725-31
The abovementioned text has also been added to the ethical considerations paragraph.
Reviewer 2 Report
- It is very interesting topic and valuable article.
- The main problem is with the methodology as - it is a systematic review and therefore the appropriate method of making it should be used as for example PRISMA.
- In all text in case of references - you should write dot after [14]. not before .[14]
- Also the idea and purpose of this second part of the article - it is not explained clearly;
Author Response
Reviewer 2
It is a very interesting topic and a valuable article.
Thank you very much for your positive comment!
The main problem is with the methodology as - it is a systematic review and therefore the appropriate method of making it should be used as for example PRISMA.
We agree, as no data can be compared due to the high heterogeneity of the included reports, this review is not a systematic review per se and according to the Cochrane Collaboration or PRISMA guidelines. Nevertheless we tried to systemize our data acquisition, quality assessment and evaluation as close to PRISMA as possible. Therefore we erased the wording “systematic” from the title of our manuscript.
In all text in case of references - you should write dot after [14]. not before .[14]
We have changed the references accordingly.
Also the idea and purpose of this second part of the article - it is not explained clearly;
Unfortunately, we do not quite understand your definition of the “second part”.
We hope to have answered all your questions and to have revised our manuscript according to your suggestions and satisfaction.
We would be more than happy if you accept our manuscript for publication in your journal.
Sincerely,
Robert Bergholz
Round 2
Reviewer 1 Report
Thank you for the extensive revision of the manuscript. I am happy with the changes, and recommend publikation in the present form.
Reviewer 2 Report
Dear Author,
Regarding to my previous comment "Also the idea and purpose of this second part of the article - it is not explained clearly", under the meaning of "second part of article" I meant this part after the references; however in the present version there is not material / text after the references; - so, it is ok now.
The correction of goal / hypothesis and deleting the word systematic improved the quality of the paper.